# Pulse Consumption and Metabolic Syndrome: Findings from the Hispanic Community Health Study/Study of Latinos

**DOI:** 10.3390/nu17213392

**Published:** 2025-10-29

**Authors:** Juliana Teruel Camargo, Gabriela Recinos, Amanda S. Hinerman, Chelsea Duong, Erik J. Rodriquez, Jordan J. Juarez, Amanda C. McClain, Sarah K. Alver, Martha L. Daviglus, Linda Van Horn, Eliseo J. Pérez-Stable

**Affiliations:** 1Division of Intramural Research, National Institute on Minority Health and Health Disparities, National Institutes of Health, Bethesda, MD 20892, USA; gabriela.recinos@nih.gov; 2Epidemiology and Community Health Branch, Division of Intramural Research, National Heart, Lung, and Blood Institute, National Institutes of Health, Bethesda, MD 20892, USA; amanda.hinerman@nih.gov (A.S.H.); chelsea.duong@ucsf.edu (C.D.); jordanjuarez@med-net.ucla.edu (J.J.J.); eliseojps2@gmail.com (E.J.P.-S.); 3School of Exercise and Nutritional Sciences, College of Health and Human Services, San Diego State University, San Diego, CA 92182, USA; amcclain@sdsu.edu; 4Epidemiology Program, Public Health Sciences Division, Fred Hutch Cancer Center, Seattle, WA 98109, USA; salver@fredhutch.org; 5Institute for Minority Health Research, College of Medicine, University of Illinois Chicago, Chicago, IL 60612, USA; daviglus@uic.edu; 6Department of Preventive Medicine, Feinberg School of Medicine, Northwestern University, Chicago, IL 60611, USA; lvanhorn@northwestern.edu

**Keywords:** Hispanic or Latino, metabolic syndrome, diet, *Fabaceae*, *Phaseolus*, lens plant, *Cicer*

## Abstract

**Background/Objectives**: Metabolic syndrome affects half of middle-aged (ages 45–64) Hispanic or Latino (Latino) adults. Pulses, fiber-rich plant proteins common in Latino diets (e.g., dry beans and lentils), may mitigate metabolic syndrome. We evaluated the association between pulse intake and metabolic syndrome. **Methods**: We analyzed data from 6,958 adults aged ≥ 50 in the Hispanic Community Health Study/Study of Latinos (2008–2011) Visit 1. Pulse intake was assessed using two 24 h dietary recalls and categorized into no, low (<1/2 cup), moderate (≥1/2 to 3/4 cup), and high pulse (>3/4 cup) daily intake groups. Metabolic syndrome was defined by criteria including blood pressure ≥130/85 mmHg or medication use, triglycerides ≥150 mg/dL or medication use, high-density lipoprotein cholesterol (men <40 mg/dL and women <50 mg/dL), and waist circumference (men ≥102 cm and women ≥88 cm). We used multivariate logistic regression models with predicted probability proportions to assess the association adjusted for sociodemographic factors, acculturation, diet quality, energy intake, and physical activity. **Results**: Of the 6,958 participants, 53.1% had metabolic syndrome and 53.4% had a moderate or high pulse intake. Pulse intake varied, where 19.4% had a high intake, 33.9% had a moderate intake, 12.5% had a low intake, and 34.2% had no intake. Moderate (predicted marginal = 0.52, 95% confidence interval [CI] = 0.49, 0.55) and high (predicted marginal = 0.49, 95%CI = 0.45, 0.53) intakes were associated with a lower prevalence of metabolic syndrome. **Conclusions**: Among Latino adults ≥50 years old, a moderate or high pulse intake was associated with a lower prevalence of metabolic syndrome. Increasing the pulse intake in the population may be linked to reduced metabolic syndrome.

## 1. Introduction

One in two middle-aged Hispanic or Latino (Latino) adults in the U.S. meets the criteria for metabolic syndrome, and three of the top five causes of death in this population are associated with metabolic syndrome [1,2,3]. Among Latino adults, metabolic syndrome has a higher prevalence among those who are older, female, of Puerto Rican heritage, and more acculturated [1,4]. Metabolic syndrome consists of a cluster of conditions that collectively increase the risk of developing heart disease, stroke, and type 2 diabetes [5,6,7]. For a diagnosis, a person must exhibit at least three out of the five following risk factors: abdominal obesity, hypertriglyceridemia, impaired glucose metabolism, elevated blood pressure, and low levels of high-density lipoprotein cholesterol [5].

Diets high in fiber and anti-inflammatory components are a key intervention for preventing and managing metabolic syndrome [8]. Worldwide, nine types of legume crops are consumed: five are pulses (chickpeas, cowpeas, dry beans, dry peas, and lentils), two are undried legumes (snap beans and snap peas), and two are oilseed legumes (peanuts and soybeans). However, the nutritional profile and benefits of legumes vary substantially. For example, non-oilseed legumes (pulses and undried legumes) are usually lower in fat and higher in fiber, while oilseed legumes are higher in fat and lower in fiber [9]. Pulses, the dried seeds of the legume family, are rich in fiber and bioactive anti-inflammatory substances such as phenols, polysaccharides, and peptides [10]. Previous studies have shown that consuming one half-cup (113 g) of pulses daily was associated with reduced postprandial hyperglycemia, lowered total and low-density lipoprotein cholesterol, increased satiety, and promoted weight loss [10,11]. Health-promoting diets such as the Mediterranean Diet, Dietary Approaches to Stop Hypertension (DASH), and plant-based diets incorporate pulses as a key source of protein, which help in the management of blood pressure, diabetes, weight, and triglyceride levels (all components of metabolic syndrome) [8].

Although the diet of the Latino population has not been extensively studied, research indicates that diet quality varies significantly across different Latino heritage groups [12]. Overall, diets in this population tend to receive high scores for nuts and legume consumption [13]. Many traditional Latin American cuisines include pulses, such as beans, which are a staple across the region [14]. Yet, pulse consumption varies greatly by country, with high levels observed in Nicaragua (25.1 kg per capita per year), Brazil (16.1 kg per capita per year), and Mexico (12.6 kg per capita per year), and with much lower levels in Chile (3.0 kg per capita per year) and Argentina (1.1 kg per capita per year) [14]. Despite the existing evidence of the health benefits of consuming pulses and their cultural significance for the Latino population, this food group remains underrecognized and miscategorized in the U.S. Using data from the National Health and Nutrition Examination Study (NHANES), one study found that only 7.9% of the U.S. adult population consumes pulses daily, with amounts ranging from 23.3 to 277.1 g/day. Notably, the Latino population has been reported to have the highest proportion of daily consumption [11]. In the U.S., terms such as legumes, beans, and pulses are often used interchangeably and do not adhere to the distinct international definitions [15]. Pulses are rich in proteins, fibers, and other biocomponents, necessitating clearer terminology, consistent global definitions, and well-defined recommendations in the Dietary Guidelines for Americans [9]. Currently, the guidelines are unclear, as pulses are sometimes categorized with vegetables and, at other times, considered part of the protein food group [16]. This lack of clear nomenclature and harmonization with international references contributes to the vague dietary recommendations for pulses in the U.S.

The main goal of this research was to assess the association between pulse consumption and the prevalence of metabolic syndrome in U.S. Latino adults. Our secondary aim was to describe pulse consumption by sociodemographic and lifestyle factors, anthropometric measures, and metabolic syndrome by Latino heritage groups. Our rationale for examining differences based on Latino heritage groups is based on differences in food group consumption and metabolic syndrome prevalence among heritage groups previously reported in the Hispanic Community Health Study/Study of Latinos (HCHS/SOL) [1,17]. Overall, insights gained could further inform targets for promoting pulse consumption among Latino adults across the U.S.

## 2. Materials and Methods

### 2.1. Study Design and Participants

The Hispanic Community Health Study/Study of Latinos (HCHS/SOL) is a multicenter, population-based cohort of 16,415 self-identified Latino adults aged 18–74 years. Participants were selected through multistage probability sampling from census block groups in four U.S. cities: Bronx, NY; Chicago, IL; Miami, FL; and San Diego, CA. Baseline assessments conducted between 2008 and 2011 included comprehensive interviews and clinical evaluations. Anthropometric measurements (height, weight, and waist circumference) and fasting blood samples were collected by trained study staff. Dietary data were collected via two 24 h dietary recalls (further details below). This study was approved by the Institutional Review Boards at all participating study centers, and all participants provided written informed consent [18].

In this study, we conducted a cross-sectional analysis since dietary data for the HCHS/SOL are only available at baseline. Of the 16,415 individuals at Visit 1, we excluded those with missing dietary information (*n* = 86) and with implausible energy intake (defined as <500 or >6000 kcal/day for females and <800 or >6000 kcal/day for males; *n* = 234) [18]. We excluded participants younger than 50 years old (*n* = 9137) because of the very low prevalence of metabolic syndrome in this group [1,2]. This resulted in a final analytic sample of 6958 adults.

### 2.2. Metabolic Syndrome Definition

Metabolic syndrome was assessed as a binary variable (yes/no) based on the National Cholesterol Education Program—The Adult Treat Panel III criteria [19], harmonized with self-reported medication use. A diagnosis required the presence of three or more of the following: (1) systolic/diastolic blood pressure ≥130/85 mmHg or blood-pressure-lowering medication use; (2) triglycerides ≥150 mg/dL or lipid-lowering medication use; (3) HDL cholesterol <40 mg/dL for men and <50 mg/dL for women; (4) fasting plasma glucose ≥100 mg/dL or glucose-lowering medication use; and (5) waist circumference ≥102 cm for men and ≥88 cm for women. Blood pressure was measured three times at 1 min intervals following a 5 min seated rest, using an automatic sphygmomanometer (Omron model HEM-907 XL, Omron Healthcare Inc., Bannockburn, IL, USA). The average of the three readings was used. Triglyceride, HDL-C, and fasting plasma glucose values were log-transformed for normalization. Medication use in the month before the baseline visit was recorded via standardized interviews and, when available, confirmed through barcode scans or manual coding of prescriptions. Waist circumference was measured at the uppermost lateral border of the right ilium to the nearest 0.1 cm.

### 2.3. Dietary Assessment and Estimation of Daily Dietary Pulse Consumption

Dietary intake was assessed using two non-consecutive 24 h dietary recalls, conducted by trained interviewers in the participant’s preferred language. The first recall was administered in person at the baseline visit, and the second recall was either administered in person or via telephone, at least 5 days or at most 45 days post-visit [12,17]. This recall captured comprehensive details on all foods, beverages, and dietary supplements consumed from midnight to midnight the previous day [20]. The primary analytical sample included participants from all heritages across the four study sites who had at least one reliable 24 h dietary recall at baseline. Participants who did not have dietary recall data were excluded from the analysis (*n* = 86).

Pulse consumption was assessed from the two 24 h dietary recalls through the food group report derived from the Nutritional Data System Research (NDSR) dietary analysis program. The food group report includes specific data on single and mixed dishes, which allowed us to identify foods that included pulses. We selected single and mixed dishes for analysis, which included chickpeas, cowpeas, dry beans, dry peas, and lentils. To account for dietary measurement error and estimate usual intake distribution, the National Cancer Institute (NCI) method was applied as described in previous publications [21,22]. One serving of pulses was defined as ½ cup (113.4 g) [23]. Daily pulse consumption was categorized based on sample distribution: non-consumer, low (<½ cup), moderate (≥½ to ¾ cup), and high (>¾ cup).

Usual daily energy intake was estimated from 24 h dietary recalls using the NCI method to adjust for measurement error [21,22]. This residual adjustment method involves statistically controlling for food intake to account for energy intake through regression analysis [24]. Diet quality was assessed using two scoring criteria: the Alternative Heathy Eating Index (AHEI)-2010 [25] and a score designed based on the Dietary Approaches to Stop Hypertension (DASH) eating pattern [26]. The NCI method was employed to estimate the usual intake amounts for each AHEI-2010 component derived from 24 h dietary recalls [24]. Each dietary component was scored from 0 to 10, reflecting minimal to maximal adherence to recommended intake levels. The AHEI-2010 comprises six components where a higher intake is considered healthiest (vegetables without potatoes, whole grains, whole fruits without fruit juice, nuts and legumes, eicosapentaenoic acid (EPA) and docosahexaenoic acid (DHA), and polyunsaturated fatty acids (PUFAs)), one component where a low-to-moderate intake is considered the healthiest (alcohol), and four components where avoidance or minimal intake is considered the healthiest (sugar-sweetened beverages and fruit juices, red and processed meats, trans fat, and sodium). Recommendations for whole grains and alcohol intake differ based on sex. The total AHEI-2010 score was calculated as the sum of the 11 dietary component scores, ranging from 0 to 110, with higher scores indicating a healthier diet quality based on the adherence to recommended intake levels [25]. The DASH scoring was assessed using 24 h dietary recalls, where participants were assigned a score from 1 to 5 based on sex-specific quintiles of saturated fatty acids, potassium, calcium, and fiber (with 5 being the most favorable). A score at or above the 60th percentile was classified as a healthier diet [27].

### 2.4. Sociodemographic and Lifestyle Factors

Age was calculated in years based on the participant’s date of birth and clinic visit date. Sex was recorded as a categorical variable (female or male). Educational attainment was categorized into three groups: no high school diploma or General Education Development (GED) credential, a high school diploma or GED, or greater than high school or GED education. Income was measured using the household income-to-poverty ratio in U.S. dollars, grouped into four categories: <100%, 101–200%, 201–300%, and >300% of the poverty threshold.

Latino heritage was categorized into Central American, Cuban, Dominican, Mexican, Puerto Rican, South American, and more than one or other heritage using a derived variable. U.S.-born status was determined by a variable that categorized participants as either born in the 50 U.S. states and the District of Columbia (DC), not including U.S. territories, or not. The language subscale of the Short Acculturation Scale for Hispanics was used to assess language acculturation [28]. This scale is scored from 1 (least acculturated) to 5 (most acculturated). Based on sample distribution, participants with scores <2 were considered less acculturated, while those with scores ≥2 were classified as more acculturated. Health insurance coverage of any type at the time of the baseline visit was recorded as a categorical variable (no or yes).

Physical activity levels were measured using the Global Physical Activity questionnaire which accounts for sedentary behavior and activity across work, travel, and leisure domains. Total physical activity was reported as the average minutes per day [29]. Cigarette use was categorized as current (every day or some days), former, or never.

### 2.5. Anthropometric Factors

Body Mass Index (BMI) was calculated from clinical measurements of weight and height using the following formula: weight (kg) divided by height squared (m^2^). BMI was then categorized according to World Health Organization criteria: underweight (BMI < 18.5), normal (18.5 ≤ BMI < 25), overweight (25 ≤ BMI < 30), and obesity (BMI ≥ 30) [30]. Abdominal obesity was assessed using the clinical waist circumference measurement in centimeters. Later, waist circumference was categorized as abdominal obesity according to the Adult Treat Panel III of the National Cholesterol Education Program criteria: ≥102 cm (male) and ≥88 cm (female) [31].

### 2.6. Statistical Analysis

To characterize pulse consumption within the population, our initial step involved quantifying the amount of pulse consumed. This was achieved through an analysis of dietary intake data obtained from 24 h dietary recalls, which were adjusted for measurement error to ensure precision, as previously described in the dietary assessment and estimation section above. Subsequently, we categorized pulse consumption based on the distribution of intake within the population in serving sizes, where 1 serving size is ½ cup (113.4 g). This approach was chosen to facilitate the identification of differences in consumption levels, given that the Dietary Guidelines for Americans do not specify recommended amounts for pulse consumption. By employing this categorization, we aimed to provide a detailed understanding of consumption patterns in the absence of established guidelines.

Descriptive statistics were used to compare sociodemographic, lifestyle, anthropometric, and medication use factors with pulse consumption categories. For continuous variables age, energy intake, and total physical activity, weighted means and standard deviations were calculated. For all the other sociodemographic, lifestyle, and anthropometric categorical variables, we analyzed by reporting frequencies and weighted percentages. Logistic regression was used to assess differences in sociodemographic, lifestyle, and anthropometric factors across the pulse intake categories.

Descriptive statistics were also used to compare metabolic syndrome diagnosis by pulse intake categories. Pulse intake categories were analyzed by reporting frequencies and weighted percentages. Differences in metabolic syndrome by pulse consumption categories were assessed using logistic regression.

Logistic regression models evaluated the possible benefits of a moderate and high consumption of daily pulse consumption for the total sample. Bivariate models were constructed to examine the relationship between the consumption level of pulse categories (low intake vs. moderate or high intake) and the outcome of interest (metabolic syndrome). Next, a multivariate logistic regression model was used to assess the association between pulse consumption and metabolic syndrome cross-sectionally with covariate adjustment. The multivariate logistic regression models included participants with non-missing data for pulse consumption, covariates, and outcome data. Covariates were selected utilizing the literature and prior knowledge (study site, Latino heritage, sex, birthplace, educational attainment, cigarette use, physical activity, age, acculturation, and energy intake) [32,33,34]. We included energy intake and diet quality in our analysis of pulses for two reasons: (1) Understanding the total caloric consumption is essential for interpreting overall dietary patterns. Pulses are nutrient-dense foods that require consideration of the total energy they add to the diet [34]. (2) Metabolic syndrome is affected by a variety of other dietary factors (e.g., sugar-sweetened beverages), and by evaluating diet quality, we can assess how the inclusion of pulses, which are known for favorable nutrient profiles, enhances overall diet quality and contributes to metabolic outcomes [25]. We reported the predicted marginal proportion for the results of these models for each level of pulse consumption and 95% confidence limits. The models were repeated, controlling for diet quality as measured by either the AHEI or the DASH. The variance inflation factor (VIF) was used to assess multicollinearity between model covariates of the level of pulse consumption, energy intake, and, when applicable, diet measures. For the reported models, pulse consumption, energy intake, and diet measure all had a VIF of less than 3. A VIF < 3 indicates a low correlation between the variables in the model. We also performed exploratory analysis for Latino heritage groups based on prior knowledge of differences by heritage groups. Survey analysis procedures were used to account for complex sampling and weighting. All analyses were conducted using SAS, Version 9.4 (SAS Institute Inc., Cary, NC, USA).

## 3. Results

### 3.1. Sociodemographic, Lifestyle, and Anthropometric Factors

Among the 6958 participants included (ages ≥50 years), the average age was ~60 years old, and the majority were female (56.1%). Socioeconomic status was generally low, with 43.1% of participants reporting no high school diploma or GED, 74.1% reporting annual household incomes at or below 200% of the federal poverty level, and 38.3% not having health insurance coverage. Most participants (92.6%) were immigrants, and 72.2% were classified as less acculturated. The three most represented Latino heritages included Mexican (29.2%), Cuban (28.8%), and Puerto Rican (19.2%) (Table 1).

In terms of lifestyle factors, the average AHEI diet quality score among the population was 50.7 out of a possible 110 points. Participants demonstrated healthier-than-average consumption patterns for nuts and legumes and polyunsaturated fatty acids, alongside a reduced intake of trans fat and sodium. However, intake levels for vegetables, whole grains, whole fruits, EPA, and DHA were below recommended thresholds. In contrast, the consumption of sugar-sweetened beverages, fruit juices, alcohol, and red and processed meats exceeded the recommended levels. More than half of participants (52.7%) reported a DASH diet quality as healthy. However, those who did not consume pulses or had a low daily intake had a higher percentage of not following a healthy DASH diet (56.5% and 66.1%, respectively). On average, participants engaged in 95.7 min of physical activity daily, spanning work, transportation, and recreation activities. Over half of the participants (53.7%) had never smoked. Regarding body weight, 83.8% of participants were categorized as having excessive weight (overweight or obesity), and 66.6% had abdominal obesity (Table 2).

### 3.2. Daily Pulse Consumption

Two thirds of the participants (*n* = 4584) reported a daily consumption of pulses, with the most common consumption level being moderate (51.5%). Participants who consumed pulses at moderate levels exhibited a statistically significantly better diet quality, with an average score of 52.6 out of 110, and had a statistically significantly healthier intake of whole grains and whole fruits compared to other consumer groups (Table 2). This group also had a statistically significantly higher proportion of never smokers (61.7%) and individuals with lower educational attainment (48.2%) (Table 1). Participants with the statistically significantly lowest diet quality scores were those with a low daily pulse consumption (48.7) rather than those who did not consume pulses daily (49.4). Those who did not consume pulses exhibited statistically significantly better consumption patterns for vegetables (4.1 vs. 3.2), whole grains (3.1 vs. 2.8), whole fruits (2.5 vs. 2.2), and EPA and DHA (3.5 vs. 2.9) compared to those with low pulse consumption.

Participants with low daily pulse consumption had the lowest diet quality score among all groups (48.7), characterized by a particularly low intake of vegetables, whole grains, and EPA and DHA, coupled with higher consumption of red and processed meats, sodium, and alcohol. Despite having the lowest caloric intake and highest total physical activity minutes, this group had the highest proportions of obesity (51.5%) and abdominal obesity (77.9%). The low consumption group comprised higher proportions of females, individuals with a high school diploma, U.S.-born participants, those more acculturated, and those with health insurance coverage compared to other groups (Table 1). Statistically significant variations in daily pulse consumption were observed across Latino heritages, with approximately 60% of Cubans reporting a high daily pulse consumption, compared to 17.1% of Mexicans and 4.4% of Puerto Ricans (Table 1).

### 3.3. Daily Pulse Consumption and Metabolic Syndrome/Latino Heritage

In this population, over 50% of participants had metabolic syndrome, with limited variation by heritage. By Latino heritage, the prevalence of metabolic syndrome was highest among Cubans (57.0%) and Dominicans (56.7%) and lowest among Mexicans (51.7%). In the overall population, those with low daily pulse consumption exhibited a higher metabolic syndrome prevalence (62%), than those with a high (51.2%) or moderate (51.3%) daily pulse consumption. This trend was consistent across Latino heritage, except among Central Americans (Figure 1).

### 3.4. Association Between Daily Pulse Consumption and Metabolic Syndrome

Adjusted weighted predicted probability models accounting for study site, heritage, sex, birthplace, education, cigarette use, physical activity, health insurance coverage, age, acculturation, diet quality, and energy intake tested the association of pulse consumption with metabolic syndrome.

The adjusted models revealed that participants with a low daily pulse consumption had the highest prevalence of metabolic syndrome (predicted marginal for all adjusted models = 0.63, 95% confidence interval [CI] = 0.58, 0.68). The second highest metabolic syndrome prevalence was observed among those with no consumption (predicted marginal model 2 = 0.54, 95%CI = 0.50, 0.58; predicted marginal model 3 = 0.53, 95%CI = 0.50, 0.57; predicted marginal model 4 = 0.52, 95%CI = 0.50, 0.58). Participants with a moderate or high daily pulse consumption had a lower prevalence of metabolic syndrome. Those with high consumption reported the lowest prevalence of metabolic syndrome in all adjusted models (predicted marginal model 2 = 0.49, 95%CI = 0.45, 0.54; predicted marginal model 3 = 0.50, 95%CI = 0.45, 0.54; predicted marginal model 4 = 0.49, 95%CI = 0.45, 0.53) (Table 3).

Analysis by heritage showed that adults of Cuban heritage who had a high consumption of pulses had a statistically significantly lower metabolic syndrome prevalence (predicted marginal model = 0.55, 95%CI = 0.50, 0.59) than those who had a low consumption (predicted marginal model = 0.77, 95%CI = 0.66, 0.86). Central Americans differed, showing a higher prevalence of metabolic syndrome at a moderate pulse consumption, but the prevalence was not statistically significant. The other heritage groups reported a lower prevalence of metabolic syndrome at a moderate or high pulse consumption, but they were not statistically significant (Table A1).

## 4. Discussion

In this study of Latino adults from six different heritages, an inverse relationship between moderate and high pulse consumption levels and the prevalence of metabolic syndrome was found. This association is supported by previous metabolic studies showing the physiological benefits of a higher pulse consumption. The potential implications for the U.S. Latino population may be considerable if the traditional high-pulse diet is promoted as healthy, consumption increases or is maintained, and potential clinical benefits are derived.

There are some inconsistencies in the results since those with a low daily pulse consumption had a higher prevalence of metabolic syndrome compared to individuals with no daily pulse consumption. This difference may be due to a healthier diet quality observed among those who did not consume pulses versus those who had a lower consumption. Those who did not consume pulses had a higher intake of fruits, vegetables, and whole grains and a lower intake of alcohol compared to those who had a low consumption. These findings suggest that future studies are needed to explore dietary patterns with or without pulses associated with metabolic syndrome, as well as the longitudinal impact of different levels of daily pulse consumption among Latino individuals on developing metabolic syndrome.

Consistent with our findings, a study on Costa Rican adults demonstrated that a 2:1 consumption ratio of beans to white rice was linked to improved cardiovascular risk factors, and substituting one serving of white rice with beans reduced the odds of developing metabolic syndrome [13]. Additionally, a randomized clinical trial involving adults with metabolic syndrome showed that those consuming a breakfast including a ½ cup of black beans had a better insulin response five hours after the meal compared to those without beans [35]. A plausible biological mechanism by which pulses might influence metabolic syndrome is due to their unique bioactive compounds (e.g., polyphenols, phytosterols), fiber, and protein content, as well as their prebiotic-rich nature. An animal experimental study using seed extract from pulses found that pulse seed extract effectively reduced cholesterol and insulin levels by 80% and decreased protein carbonylation, a key process in oxidative stress [36]. A systematic review reported that the resistance starch found in beans and pulses has the potential to improve metabolic health by fostering the growth of beneficial gut bacteria, specifically from four major phyla, Firmicutes, Bacteroidetes, Proteobacteria, and Actinobacteria, and by enhancing the production of short-chain fatty acids in the colon [37]. Further research is needed to understand the impact of pulse seed oils in humans and whether the sources of the seeds could impact the biological mechanisms of metabolic syndrome, as well as the effects of pulses on gut microbiota and metabolic syndrome.

We also observed that a higher daily pulse consumption was linked to a better diet quality and a trend of lowering obesity, despite also being linked to a higher caloric intake. Even though our findings were observed in unadjusted models, they are similar to findings from previous randomized clinical trials, which showed that increased pulse consumption was associated with weight loss, even without calorie restriction [38,39]. Diets incorporating pulses resulted in more significant weight loss compared to those without pulses, regardless of whether caloric restrictions were in place [40]. Conversely, participants in our population who had a low daily pulse consumption tended to exhibit a poorer diet quality and higher obesity and metabolic syndrome. Pulses also play a significant role in weight management, largely because of their high fiber and protein levels, which help inhibit starch digestion and slow gastric emptying. This leads to increased satiety and reduced food intake [40].

The average pulse consumption in our study was much higher (114.5 g per day) compared to other U.S. cohorts that did not include Latinos or had lower percentages than the national average, which varied from 2 to 84 g per day [41]. Overall, the average daily pulse consumption was comparable to that observed in Latin America (140 g per day) [41] and aligns with previous research on Puerto Rican dietary habits [42]. These findings suggest that participants in our study might be following a pulse consumption pattern more similar to that of Latin America and the island of Puerto Rico rather than the 50 U.S. states, as expected since most were foreign-born and less acculturated. Moreover, there was unexpected variation within a heritage group. For example, among Cubans, 33% did not consume pulses daily, while 49% had a high daily consumption. This indicates that factors beyond cultural heritage, such as individual preferences [42], access to food and ingredients [4], birthplace, acculturation [27,43], and others might influence pulse consumption and need further research.

We found an association between high pulse consumption and odds of metabolic syndrome among participants of Cuban heritage. A previous study of dietary intakes in the HCHS/SOL highlighted cultural differences in eating habits among Hispanic/Latino adults. Specifically, Cubans tend to have higher intakes of total energy, refined grains, vegetables, red meat, and fats. In contrast, Mexicans have a higher consumption of whole grains, while Puerto Ricans exhibit lower intakes of fruits and vegetables but higher consumption of sugar-sweetened beverages, total fats, and saturated fats. Dominicans had a diet higher in fruits and poultry, and Central and South Americans showed the highest intake of fruits, poultry, and fish. These significant variations in dietary habits reflect the diverse cultural heritage and countries of origin within the Hispanic/Latino population [17]. The traditional Cuban diet heavily features foods like rice, black beans, pork, plantains, and starchy vegetables such as yuca and malanga. Many traditional dishes incorporate pulses, including split peas with rice and eggs (*arroz*, *chícharos y huevos*), split pea creamy soups (*potajes con viandas*), and lentil soup with plantains (*sopa de lentejas con plátano*) [44]. These culinary practices, which involve combining pulses with vegetables, fish, and eggs, may contribute to greater health benefits.

Several dietary guidelines across Latin America and Europe emphasize the importance of pulses as part of the protein food group, highlighting their consumption for cultural, economic, and environmental reasons [45,46,47,48,49]. Pulses, particularly beans, are integral to many traditional dishes in Latin America, offer a cost-effective alternative to animal-based proteins, and represent environmentally sustainable crops [50,51]. Until 2020, the U.S. Dietary Guidelines for Americans (DGA) did not specifically recognize pulses as an important food group to be emphasized. In the forthcoming 2025 DGA, the health benefits of pulses are acknowledged under the categories of beans, peas, and lentils. However, pulses are not identified as a distinct component requiring specific recommendations; they remain categorized within the broader protein and vegetable recommendations [16]. The long-standing terminological confusion between “legumes” and “pulses” in the U.S. can lead to misinterpretation and confusion among consumers, health professionals, and researchers [11]. Adopting the term “pulses” could reduce ambiguity, thereby strengthening recommendations and research related to this unique class of legumes, which offer significant nutritional and health benefits [9] and hold cultural relevance for specific population groups, such as Latino communities.

Pulses are already an integral part of Latino gastronomy, and their increased consumption could bring significant health benefits to this population. Previous studies have shown that Latino individuals generally have a positive attitude toward consuming pulses. Increased efforts in educating this population, particularly U.S.-born individuals, about the health benefits and cooking methods of pulses could further promote their consumption [43]. At a structural level, enhancing accessibility by addressing price variations and clearly labeling the sodium content of canned pulses could significantly benefit consumers [43]. Furthermore, incorporating clear recommendations for pulse consumption in the U.S. Dietary Guidelines would support cardiometabolic health for the U.S. population.

This study has several limitations. Being cross-sectional in design, it cannot inherently establish temporality. An alternative explanation would also be reverse causality, as people diagnosed with metabolic syndrome could have changed their diet, either increasing or decreasing their pulse consumption. This study identifies opportunities for further research on causality and reverse causality for pulses and metabolic syndrome. Participants’ dietary intake was self-reported. However, using multiple 24 h dietary recalls is considered the gold standard for assessing individual dietary intake [20], and using the NCI method to estimate the distribution of pulse intake, correcting for daily intake error variance and accounting for non-dietary variance, added rigor to our methods [52]. Lastly, this study may have confounders and other factors associated with eating that may not have been captured in the present analysis, which could be explored in future studies.

## 5. Conclusions

In conclusion, promoting an increased pulse intake among U.S. Latino communities represents a culturally tailored, low-cost intervention with potentially significant benefits for cardiometabolic health. Future research is needed to investigate the long-term and causal relationships between pulse consumption and cardiovascular health outcomes. Further exploration of socioeconomic and culturally specific dietary practices that may influence these relationships may help in tailoring personalized dietary recommendations.

## Figures and Tables

**Figure 1 nutrients-17-03392-f001:**
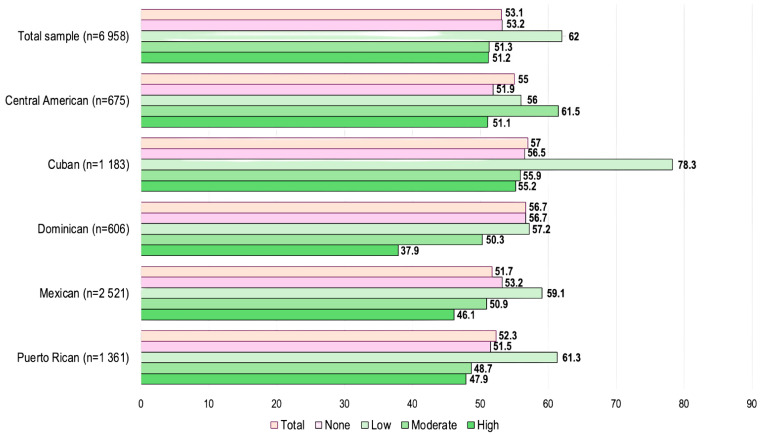
Baseline metabolic syndrome prevalence by category of daily pulse ^1^ consumption among participants 50 years and older, Hispanic Community Health Study/Study of Latinos, 2008–2011 (*n* = 6958). (^1^ Pulses included chickpeas, cowpeas, dry beans, dry peas, and lentils. Notes: Low = <1/2 cup; Moderate = ≥1/2 to 3/4 cup; High = >3/4 cup).

**Table 1 nutrients-17-03392-t001:** Weighted baseline sociodemographic characteristics by category of daily pulse ^1^ consumption among participants 50 years and older, Hispanic Community Health Study/Study of Latinos, 2008–2011 (*n* = 6958).

Characteristics		Total Sample (*n* = 6958)	Non-Consumer (*n* = 2374)	Low (<1/2 Cup) (*n* = 869)	Moderate (≥1/2 to 3/4 Cup) (*n* = 2361)	High (>3/4 Cup) (*n* = 1354)	*p*
Age, years—mean (SD)		59.6 (0.1)	59.4 (0.2)	58.3 (0.4)	58.9 (0.3)	61.4 (0.3)	<0.001
Sex *n* (%)	Female	4364 (56.1)	1555 (58.3)	721 (79.8)	1799 (69.3)	289 (26.5)	<0.001
	Male	2594 (43.9)	819 (41.7)	148 (20.2)	562 (30.7)	1065 (73.5)	
Educational attainment *n* (%)	No high school diploma or GED	3253 (43.1)	1009 (40.1)	391 (39.7)	1224 (48.2)	629 (43.0)	0.002
	High school diploma or GED	1360 (19.0)	489 (18.6)	187 (25.0)	435 (18.8)	249 (17.4)	
	Greater than high school or GED	2317 (37.9)	870 (41.2)	290 (35.2)	682 (33.1)	475 (39.6)	
Annual household income by federal poverty level *n* (%)	≤100%	2539 (39.9)	810 (41.2)	326 (38.7)	889 (42.4)	513 (42.9)	0.01
	101 to 200%	2275 (34.2)	797 (33.8)	255 (31.4)	777 (33.1)	446 (37.4)	
	201 to 300%	846 (14.1)	318 (15.1)	100 (16.4)	274 (13.8)	154 (12.0)	
	>300%	614 (11.8)	246 (14.9)	94 (13.5)	179 (10.7)	95 (7.7)	
Study site *n* (%)	Bronx	1704 (26.9)	595 (28.4)	391 (48.2)	584 (31.6)	134 (10.0)	<0.001
	Chicago	1663 (12.2)	489 (11.0)	181 (10.1)	619 (13.7)	374 (13.2)	
	Miami	1885 (38.1)	650 (36.5)	157 (24.8)	356 (20.3)	722 (67.6)	
	San Diego	1706 (22.7)	640 (24.1)	140 (16.9)	802 (34.5)	124 (9.2)	
Hispanic/Latino Heritage *n* (%)	Central American	675 (6.4)	203 (5.2)	62 (5.5)	237 (6.9)	173 (7.9)	<0.001
	Cuban	1183 (28.8)	394 (27.1)	72 (14.9)	128 (9.8)	589 (59.7)	
	Dominican	606 (8.9)	154 (6.0)	73 (9.4)	240 (11.4)	139 (9.8)	
	Mexican	2521 (29.2)	843 (29.9)	188 (19.2)	1140 (42.3)	350 (17.1)	
	Puerto Rican	1361 (19.2)	526 (22.9)	315 (34.2)	428 (21.6)	92 (4.4)	
	South American	466 (5.4)	198 (6.8)	135 (11.7)	133 (5.9)	0 (0.0)	
	More than one/Other heritage	126 (2.2)	49 (2.2)	22 (5.1)	44 (2.1)	11 (1.0)	
Birthplace *n* (%)	Not born in 50 US States or DC	6376 (92.6)	2104 (90.3)	758 (87.9)	2195 (92.9)	1319 (97.9)	<0.001
	Born in 50 US States or DC ^2^	568 (7.4)	266 (9.7)	110 (12.1)	157 (7.1)	35 (2.1)	
Acculturation ^3^—mean (SD)		1.7 (0.03)	1.9 (0.05)	1.9 (0.05)	1.7 (0.03)	1.4 (0.03)	<0.001
Acculturation ^3^ *n* (%)	Less acculturated ^4^	5032 (72.2)	1583 (65.5)	558 (64.9)	1763 (72.5)	1128 (85.0)	<0.001
	More acculturated ^4^	1904 (27.8)	781 (34.5)	307 (35.1)	593 (27.5)	223 (15.0)	
Health insurance coverage *n* (%)	No	2834 (38.3)	921 (35.6)	282 (31.8)	993 (40.6)	638 (42.5)	<0.001
	Yes	4029 (61.7)	1417 (64.4)	571 (68.2)	1336 (59.4)	705 (57.5)	

Abbreviations: DC, District of Columbia; GED, General Educational Development; *n*, number; SD, standard deviation; US, United States of America. ^1^ Pulses included chickpeas, cowpeas, dry beans, dry peas, and lentils. ^2^ It does not include US Territories. ^3^ Acculturation: measured by the Short Acculturation Scale for Hispanics, language subscale. ^4^ Less acculturated: score <2; more acculturated: score ≥2. Notes: *p*-values are comparing differences among categories of daily pulse consumption.

**Table 2 nutrients-17-03392-t002:** Weighted baseline lifestyle and anthropometric characteristics by category of daily pulse ^1^ consumption among participants 50 years and older, Hispanic Community Health Study/Study of Latinos, 2008–2011 (*n* = 6958).

Characteristics		Total Sample (*n* = 6958)	Non-Consumer (*n* = 2374)	Low (<1/2 Cup) (*n* = 869)	Moderate (≥1/2 to 3/4 Cup) (*n* = 2361)	High (>3/4 Cup) (*n* = 1354)	*p*
Diet quality AHEI ^2^, mean (SD)	Overall score	50.7 (0.2)	49.4 (0.3)	48.7 (0.5)	52.6 (0.3)	51.1 (0.4)	<0.001
	Vegetables (without potatoes)	4.1 (0.04)	4.1 (0.1)	3.2 (0.1)	4.1 (0.1)	4.6 (0.2)	<0.001
	Whole grains	3.2 (0.04)	3.1 (0.1)	2.8 (0.1)	3.5 (0.1)	3.0 (0.1)	<0.001
	Whole fruit (without fruit juice)	2.6 (0.1)	2.5 (0.1)	2.2 (0.1)	3.2 (0.1)	2.2 (0.1)	<0.001
	Sugar-sweetened beverages and fruit juices	1.7 (0.04)	1.7 (0.1)	1.9 (0.1)	1.9 (0.1)	1.2 (0.1)	<0.001
	Nuts and legumes	6.6 (0.1)	5.2 (0.1)	5.3 (0.1)	7.0 (0.1)	8.9 (0.1)	<0.001
	Red and processed meats	4.0 (0.1)	4.1 (0.1)	5.1 (0.1)	4.7 (0.1)	2.5 (0.1)	<0.001
	Trans fat	8.3 (0.02)	8.2 (0.03)	8.2 (0.03)	8.2 (0.02)	8.5 (0.02)	<0.001
	EPA and DHA	3.5 (0.03)	3.5 (0.1)	2.9 (0.1)	3.5 (0.1)	3.7 (0.1)	<0.001
	PUFA	5.5 (0.03)	5.4 (0.04)	5.2 (0.04)	5.4 (0.03)	5.7 (0.04)	<0.001
	Sodium	6.6 (0.1)	7.0 (0.1)	7.5 (0.1)	6.8 (0.1)	5.3 (0.1)	<0.001
	Alcohol	4.8 (0.1)	4.8 (0.1)	4.3 (0.1)	4.5 (0.1)	5.4 (0.1)	<0.001
Diet quality DASH ^3^, mean (SD)	Not healthy (≤60th percentile)	3074 (47.3)	1247 (56.5)	554 (66.1)	800 (37.5)	473 (37.3)	<0.001
	Healthy (>60th percentile)	3880 (52.7)	1127 (43.5)	314 (33.9)	1559 (62.5)	880 (62.7)	<0.001
Energy intake, mean (SD), kcal/d		1837.3 (10.8)	1780.6 (15.1)	1571.1 (18.6)	1764.1 (15.3)	2124.6 (15.5)	<0.001
Total physical activity ^4^, mean (SD), min/d		95.7 (2.8)	91.4 (5.2)	107.1 (8.2)	95.5 (4.8)	97.2 (6.0)	0.40
Cigarette use *n* (%)	Never	3830 (53.7)	1263 (52.8)	500 (57.3)	1481 (61.7)	586 (43.9)	<0.001
	Former	1905 (27.6)	683 (29.0)	218 (24.9)	561 (23.6)	443 (31.7)	
	Current	1207 (18.7)	422 (18.2)	150 (17.8)	313 (14.7)	322 (24.4)	
BMI (kg/m^2^) ^5^ *n* (%)	Underweight	30 (0.6)	3 (0.2)	5 (0.5)	11 (0.7)	11 (1.0)	<0.001
	Normal	1073 (15.6)	363 (14.5)	102 (11.0)	349 (15.4)	259 (19.4)	
	Overweight	2728 (40.2)	924 (40.3)	309 (36.9)	912 (39.1)	583 (43.0)	
	Obesity	3104 (43.6)	1079 (45.0)	448 (51.5)	1080 (44.7)	497 (36.6)	
Abdominal obesity ^6^ *n* (%)	No	2196 (33.4)	731 (30.7)	187 (22.1)	619 (29.9)	659 (46.3)	<0.001
	Yes	4738 (66.6)	1634 (69.3)	678 (77.9)	1736 (70.1)	690 (53.7)	

Abbreviations: BMI, Body Mass Index; d, days; DHA: docosahexaenoic acid; EPA, eicosapentaenoic acid; kcal, kilocalories; kg, kilograms; m, meters; min, minutes; *n*, number; PUFA, polyunsaturated fatty acid; SD, standard deviation. ^1^ Pulses included chickpeas, cowpeas, dry beans, dry peas, and lentils. ^2^ Diet quality was defined based on the Alternative Healthy Eating Index (AHEI)-2010 scoring criteria. The total score was calculated as the sum of the 11 dietary components (range: 0–110), with higher scores indicating a healthier diet quality. For each dietary component, the score ranges from 0 to 110. For vegetables (without potatoes), whole grains, whole fruits, nuts, and legumes, in terms of EPA, DHA, and PUFA, the highest intake is considered the healthiest. For alcohol, low-to-moderate intake is considered the healthiest. For sugar-sweetened beverages and fruit juices, red and processed meats, *trans* fat, and sodium avoidance, the lowest intake is considered the healthiest. Whole grains and alcohol intake had sex-specific recommendations. ^3^ Diet quality was defined based on DASH scoring criteria. The total score ranges from 1 to 5 based on sex-specific quintile of saturated fatty acids, potassium, calcium, and fiber. ^4^ Physical activity was measured using the Global Physical Activity Questionnaire. ^5^ Underweight (<18.5); normal (≥18.5 to <25.0); overweight (≥25.0 to <30.0); obesity (≥30.0). ^6^ Abdominal obesity: waist circumference ≥102 cm (male) and ≥88 cm (female). Notes: *p*-values are comparing differences among categories of daily pulse consumption.

**Table 3 nutrients-17-03392-t003:** Weighted predicted probabilities from modeling prevalent metabolic syndrome for daily pulse ^1^ consumption among Latino adults aged 50 years and older, Hispanic Community Health Study/Study of Latinos, 2008–2011 (*n* = 6958).

Pulse Consumption	Model 1 Unadjusted (*n* = 6953)		Model 2 Adjusted for Energy Intake (*n* = 6805)		Model 3 Adjusted for Energy Intake and Diet Quality (AHEI) (*n* = 6801)		Model 4 Adjusted for Energy Intake and Diet Quality (DASH) (*n* = 6801)	
	Predicted Marginal (95% CI)	SE	Predicted Marginal (95% CI)	SE	Predicted Marginal (95% CI)	SE	Predicted Marginal (95% CI)	SE
None	0.53 (0.49, 0.57)	0.02	0.54 (0.50, 0.58)	0.02	0.53 (0.50, 0.57)	0.02	0.54 (0.50, 0.58)	0.02
Low (<1/2 cup)	0.62 (0.57, 0.66)	0.02	0.63 (0.58, 0.68)	0.03	0.63 (0.58, 0.68)	0.02	0.63 (0.58, 0.68)	0.03
Moderate (≥1/2 cup to 3/4 cup)	0.51 (0.48, 0.54)	0.02	0.52 (0.49, 0.55)	0.02	0.52 (0.49, 0.56)	0.02	0.52 (0.48, 0.55)	0.02
High (>3/4 cup)	0.51 (0.48, 0.55)	0.02	0.49 (0.45, 0.54)	0.02	0.50 (0.45, 0.54)	0.02	0.49 (0.45, 0.53)	0.02
Model *p*	0.0007		0.0006		0.0009		0.0004	
Covariates	Wald F	*p*	Wald F	*p*	Wald F	*p*	Wald F	*p*
Study site			0.47	0.71	0.42	0.74	0.46	0.71
Hispanic/Latino heritage			1.78	0.10	1.32	0.24	2.04	0.06
Sex			0.00	1.00	0.39	0.53	0.12	0.72
Birthplace			0.57	0.45	0.34	0.56	0.65	0.42
Educational attainment			5.17	0.006	4.65	0.01	5.34	0.005
Cigarette use			4.12	0.02	4.61	0.01	3.93	0.02
Physical activity			21.14	<0.001	20.54	<0.001	21.35	<0.001
Health insurance coverage			0.18	0.67	0.16	0.69	0.17	0.68
Age			10.11	0.002	14.78	<0.001	8.63	0.003
Acculturation ^2^			1.25	0.26	1.15	0.29	1.27	0.26
Energy intake			6.89	0.009	9.64	0.002	9.26	0.002
Diet quality—AHEI					6.91	0.009		
Diet quality—DASH							1.27	0.26

Abbreviations: AHEI, Alternative Healthy Eating Index; CI, confidence interval; DASH, Dietary Approaches to Stop Hypertension; SE, Standard Error. ^1^ Pulses included chickpeas, cowpeas, dry beans, dry peas, and lentils. ^2^ Acculturation measured with the Short Acculturation Scale for Hispanics.

## Data Availability

Restrictions apply to the availability of these data. Data were obtained from the Hispanic Community Health Study/Study of Latinos and are available from https://sites.cscc.unc.edu/hchs/ accessed on 5 January 2023 with permission from the Hispanic Community Health Study/Study of Latinos.

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
