# Peer review of "Pulse Consumption and Metabolic Syndrome: Findings from the Hispanic Community Health Study/Study of Latinos"

_nutrients, 2025, doi:10.3390/nu17213392_

Round 1
Reviewer 1 Report
Comments and Suggestions for Authors
First important problem is the denominator inconsistency & model population. The Methods state the analytic sample is n = 6,958 (≥50 y). However Table 3 and the main adjusted models are shown for n = 4,583, which corresponds to the consumer subset (the manuscript elsewhere reports ~4,584 daily pulse consumers). It is not sufficiently explicit whether the main logistic regression was restricted to pulse consumers only (excluding non-consumers) or to the full sample. This changes the interpretation drastically. Please explicitly state the model population for each analysis and harmonize N’s in text, tables and captions.
Then, there is a provblem with the choice of reference category and exclusion of non-consumers (interpretation hazard). Table 3 uses “low (<½ cup)” as the reference. If non-consumers were excluded, the estimates (ORs) compare moderate/high consumers only to low consumers — not to non-consumers nor to the full population. This should be justified. Excluding non-consumers may introduce selection bias and limits public-health interpretation (recommendation to “increase pulses” is different if non-consumers are excluded). If analyses are restricted to consumers, present parallel models in the full sample (non-consumers included) and/or explicitly explain rationale.
Next issue, cross-sectional design → temporality and reverse causation. The study is cross-sectional (diet and components measured at baseline) and cannot establish causality. The Discussion notes this, but authors should be explicit about reverse causation as a likely alternative explanation (e.g., persons diagnosed with metabolic syndrome could have changed diet to increase or reduce pulses). Consider stronger language and sensitivity analyses (below).
A major issue is also the over-adjustment / collider risk from diet quality variable.
Diet quality (AHEI-2010) includes a nuts & legumes component. Adjusting for AHEI including legume/pulse contribution can adjust away part of the exposure (over-adjustment). Authors must either:
- Recompute AHEI excluding the nuts & legumes component (or recompute a AHEI_without_pulses score) and use that as covariate; or
- Report models with and without AHEI to show sensitivity. Explain in text why AHEI was included and how you handled overlap between exposure and covariate.
Then, the use of odds ratios with a common outcome (misleading magnitude): Metabolic syndrome prevalence ≈53% in this sample. With common outcomes, ORs can substantially overestimate relative risk. Recommend reporting prevalence ratios (PRs) (e.g., modified Poisson with robust variance) or marginal predicted prevalences (risk differences) alongside ORs. Also present absolute differences (predicted prevalence in each category). This improves interpretability and public-health messaging.
Then, the manuscript notes that the low-consumption group had worse diet quality and higher obesity than non-consumers — an unintuitive result. Please expand discussion on potential selection, measurement or residual confounding explanations, and consider analyses to explore this directly (e.g., characterize the non-consumer subgroup).
These are first line modifications that have to be carried out in order to improve the quality of the article.
Author Response
Response to Reviewer 1
Comment 1:
First important problem is the denominator inconsistency & model population. The Methods state the analytic sample is n = 6,958 (≥50 y). However, Table 3 and the main adjusted models are shown for n = 4,583, which corresponds to the consumer subset (the manuscript elsewhere reports ~4,584 daily pulse consumers). It is not sufficiently explicit whether the main logistic regression was restricted to pulse consumers only (excluding non-consumers) or to the full sample. This changes the interpretation drastically. Please explicitly state the model population for each analysis and harmonize N’s in text, tables, and captions.
Response to comment 1:
Thank you for your comment. We have updated our analysis and are now reporting predicted marginal proportions across our total sample of n = 6,958. We had previously run the logistic regression only among the consumers, which is why the sample was different. Updated methods are found on lines 249-251: " We report the predicted marginal proportion for the results of these models for each level of pulses consumption and 95% confidence limits." and updated model is shown in Table 3.
Comment 2:
Then, there is a problem with the choice of reference category and exclusion of non-consumers (interpretation hazard). Table 3 uses “low (<½ cup)” as the reference. If non-consumers were excluded, the estimates (ORs) compare moderate/high consumers only to low consumers — not to non-consumers nor to the full population. This should be justified. Excluding non-consumers may introduce selection bias and limits public-health interpretation (recommendation to “increase pulses” is different if non-consumers are excluded). If analyses are restricted to consumers, present parallel models in the full sample (non-consumers included) and/or explicitly explain rationale.
Response to comment 2:
We have updated our analysis and are now reporting predicted marginal proportions across our total sample of n = 6,958 including persons not reporting pulses consumption. The changes can be found on lines 249-251: " We report the predicted marginal proportion for the results of these models for each level of pulses consumption and 95% confidence limits." and the updated model is shown in Table 3. The results from the previous models did not change since moderate and high pulses consumption was associated with lower prevalence of metabolic syndrome.
Comment 3:
Next issue, cross-sectional design → temporality and reverse causation. The study is cross-sectional (diet and components measured at baseline) and cannot establish causality. The Discussion notes this, but authors should be explicit about reverse causation as a likely alternative explanation (e.g., persons diagnosed with metabolic syndrome could have changed diet to increase or reduce pulses). Consider stronger language and sensitivity analyses (below).
Response to comment 3:
Thank you for your suggestion. Although a formal diagnosis of “metabolic syndrome” is not often made by clinicians, individuals may have changed their diet if diagnosed with high blood pressure or abdominal obesity. The benefit of pulses consumption is not widely known, and some people may not consider it to be in the “healthy foods” category. We added on lines 486-489 further language on the possibility of reverse causality as an alternative explanation: "An alternative explanation would also be reverse causality, as people diagnosed with metabolic syndrome may have changed their diet either increasing or decreasing their pulses consumption. This study identifies opportunities for further research on causality and reverse causality on pulses and metabolic syndrome."
Comment 4:
A major issue is also the over-adjustment / collider risk from diet quality variable.
Diet quality (AHEI-2010) includes a nuts & legumes component. Adjusting for AHEI including legume/pulse contribution can adjust away part of the exposure (over-adjustment). Authors must either:
Recompute AHEI excluding the nuts & legumes component (or recompute a AHEI without pulses score) and use that as covariate; or
Report models with and without AHEI to show sensitivity. Explain in text why AHEI was included and how you handled overlap between exposure and covariate.
Response to comment 4:
We included all four models tested in the revised Table 3 (model 1: unadjusted, model 2: adjusted for covariates + energy intake, model 3: adjusted for covariates +energy intake and diet quality (AHEI), and model 4: adjusted for covariates + energy intake and diet quality (DASH). We included a model adjusted for diet quality using a score based on the DASH (Dietary Approaches to Stop Hypertension) diet that considers both macro- and micronutrients, rather than individual foods (such as nuts and legumes). We also added the Wald F and Wald F p-values for each covariate, which represent the impact of each covariate in the model. The associations remained very similar across the models.
Comment 5:
Then, the use of odds ratios with a common outcome (misleading magnitude): Metabolic syndrome prevalence ≈53% in this sample. With common outcomes, ORs can substantially overestimate relative risk. Recommend reporting prevalence ratios (PRs) (e.g., modified Poisson with robust variance) or marginal predicted prevalences (risk differences) alongside ORs. Also present absolute differences (predicted prevalence in each category). This improves interpretability and public-health messaging.
Response to comment 5:
Thank you for the suggestions. We modified the analysis and are now reporting weighted predicted marginal probabilities and thus avoid the risk of having a misleading magnitude from the use of odds ratios. Modified methods are found on lines 249-251. We also updated Table 3, the results section, and the abstract accordingly.
Comment 6:
Then, the manuscript notes that the low-consumption group had worse diet quality and higher obesity than non-consumers — an unintuitive result. Please expand discussion on potential selection, measurement or residual confounding explanations, and consider analyses to explore this directly (e.g., characterize the non-consumer subgroup).
These are first line modifications that have to be carried out in order to improve the quality of the article.
Response to comment 6:
Thank you for your comment. We agree that the finding of non-consumers of pulses having better quality diet and less obesity than low consumers unintuitive. We hypothesize that Latinos who have abandoned pulses in their diet are more likely to have adopted other forms of healthy nutrition that is not part of the traditional Latino diet. We expanded the discussion on lines 393-394: " Those who did not consume pulses had a higher intake of fruits, vegetables, and whole grains and a lower intake of alcohol compared to those who had a low consumption." We also modified the analysis for predicted probabilities adding the no consumption group to the analysis.
Reviewer 2 Report
Comments and Suggestions for Authors
The research investigates 6,958 Latino adults aged 50 and above from HCHS/SOL Visit 1 to establish whether their regular pulse consumption leads to metabolic syndrome development. The researchers used two 24-hour recall surveys to generate exposure data through the NCI method for usual intake estimation, where one serving equaled 113.4 grams or ½ cup. The researchers organized pulse consumption data into four groups which started with non-consumers and continued with three daily consumption levels: less than half cup (low), half to three-quarters cup (moderate) and more than three-quarters cup (high). The researchers applied survey weights to their models for site location and heritage background adjustments as well as sociodemographic factors and acculturation levels and physical activity and energy consumption and diet quality. The research demonstrated that moderate pulse consumption lowered metabolic syndrome risk by 0.64 (95% CI 0.49–0.82) when compared to low intake while high consumption reduced risk by 0.53 (0.38–0.73). The study provides information about how heritage influences both eating habits and metabolic health indicators in the population. The research implements dependable methods which receive complete documentation throughout the paper.
Strengths are substantial. The researchers employed multiple recall data to generate exposure measurements which achieved high precision for representing typical consumption of episodically eaten foods. The AHEI-based characterization and survey-weighted regression follow the most effective research practices. The research results demonstrate both scientific biological connections and consistent findings. The paper connects pulses to traditional Latino food customs and modern dietary recommendations which makes the research more useful for practice.
The paper needs additional details with minor modifications to achieve better precision without changing the final results. The exposure definition needs to have consistent definitions between written text and visual elements throughout the entire paper. The Methods section defines “low” as under ½ cup and “moderate” as between ½ to ¾ cup yet Figure 1 displays “Low = ½ cup” which contradicts the established Methods definitions. The figure note requires an update to match the Methods-defined cut points and researchers should verify all tables and captions use identical threshold values.
The authors need to address the inconsistency between their claim of using distribution-based quartiles for intake categories and their established serving size definitions. The manuscript uses fixed ½- and ¾-cup thresholds but the Methods section defines “low” as less than ½ cup and “moderate” as ½ to ¾ cup. You should describe your method for determining cut points and explain the reasoning behind your selection. The text should eliminate “quartiles” since fixed servings were used but researchers must show their actual cut points if they applied quartiles in the study.
Third, tighten internal counts. The research involved 6,958 participants who consumed pulses at least twice a week, with 4,584 participants in this group. The consumer-only models in Table 3 include 4,583 participants. Still, researchers need to confirm if one observation was excluded because of missing covariate data and include this exclusion information in the text. The text percentages require the correct denominators to calculate sex distribution percentages across intake categories in Table 1.
Fourth, prevent potential overadjustment. The AHEI-2010 scoring system includes nuts and legumes as a single scoring component. The inclusion of pulses within the “nuts and legumes” component of the AHEI-2010 scoring system might result in exposure-related variance being partially controlled through adjustment. The researchers should conduct a sensitivity analysis using an AHEI version that excludes nuts/legumes, or select a different diet-quality score that excludes legumes. The Methods section requires an explanation about the AHEI version used for sensitivity tests, along with descriptions of any notable outcome variations.
Fifth, report model diagnostics and trends. The researchers conducted survey-weighted logistic regression analysis and performed linear trend tests to evaluate ordered intake categories through median gram values for each group. The study should present an adjusted predicted prevalence or marginal probabilities graph that illustrates dose–response effects at various intake levels. The researchers need to check energy intake, AHEI, and pulse grams for acceptable multicollinearity and present variance inflation factors or make a brief statement about it. The researchers need to extend their analysis of participants who consumed low amounts of food. The low consumer group showed the lowest diet quality and highest obesity and abdominal obesity rates despite their reported low calorie intake and high physical activity minutes according to descriptive tables which suggest possible measurement errors or confounding factors. The paper requires additional details about energy reporting inaccuracies and NCI assessment methods for lifestyle behaviors to achieve better transparency. The study needs to conduct a sensitivity analysis that removes data to eliminate implausible energy reports.
The researchers need to organize their presentation of heritage-specific findings in a more structured way. The supplemental tables show that different heritage groups exhibit varying levels of connection between pulse consumption and metabolic syndrome risk, with particular groups showing more pronounced relationships. The main text requires a short overview of cultural eating habits and details about whether researchers tested an interaction term (p-interaction). The study establishes its research boundaries and explains how variables relate to each other in the eighth step. The Discussion section reveals that the study used a cross-sectional design and participants answered survey questions based on self-reported information. The analysis contains two statements about metabolic syndrome patients changing their bean consumption habits and uncontrolled confounding factors that include food insecurity and cooking techniques. The research maintains a relationship between variables but avoids making any claims about causality.
The tenth step requires researchers to enhance their reporting quality. The researchers need to detail their survey methods, confirm the use of weight measurements, strata, and PSU components, and describe their strategy for handling missing covariate data. The study needs to include 95% confidence intervals for all prevalence estimates in the text and complete p-values for instances where “<.001” appears in the journal's preferred format. The abstract needs to present the complete category definitions and adjusted ORs that appear in the final version.
Author Response
Comment 1:
The research investigates 6,958 Latino adults aged 50 and above from HCHS/SOL Visit 1 to establish whether their regular pulse consumption leads to metabolic syndrome development. The researchers used two 24-hour recall surveys to generate exposure data through the NCI method for usual intake estimation, where one serving equaled 113.4 grams or ½ cup. The researchers organized pulse consumption data into four groups which started with non-consumers and continued with three daily consumption levels: less than half cup (low), half to three-quarters cup (moderate) and more than three-quarters cup (high). The researchers applied survey weights to their models for site location and heritage background adjustments as well as sociodemographic factors and acculturation levels and physical activity and energy consumption and diet quality. The research demonstrated that moderate pulse consumption lowered metabolic syndrome risk by 0.64 (95% CI 0.49–0.82) when compared to low intake while high consumption reduced risk by 0.53 (0.38–0.73). The study provides information about how heritage influences both eating habits and metabolic health indicators in the population. The research implements dependable methods which receive complete documentation throughout the paper.
Strengths are substantial. The researchers employed multiple recall data to generate exposure measurements which achieved high precision for representing typical consumption of episodically eaten foods. The AHEI-based characterization and survey-weighted regression follow the most effective research practices. The research results demonstrate both scientific biological connections and consistent findings. The paper connects pulses to traditional Latino food customs and modern dietary recommendations which makes the research more useful for practice.
The paper needs additional details with minor modifications to achieve better precision without changing the final results. The exposure definition needs to have consistent definitions between written text and visual elements throughout the entire paper.
Response to comment 1:
Thank you for your overall summary of the manuscript and the study strengths comment. We reviewed the pulses consumption cut-off points throughout the text and visual elements maintained consistent definitions of the measures of being low as < 1/2 cup, moderate is ≥1/2 cup to 3/4 cup, and high is defined as > 3/4 cup.
Comment 2:
The Methods section defines “low” as under ½ cup and “moderate” as between ½ to ¾ cup yet Figure 1 displays “Low = ½ cup” which contradicts the established Methods definitions. The figure note requires an update to match the Methods-defined cut points and researchers should verify all tables and captions use identical threshold values.
Response to comment 2:
Thank you for pointing out this discrepancy which was an error. We reviewed the tables and figures and modified them to match the methods section definition: low < 1/2 cup, moderate ≥1/2 cup to 3/4 cup, and high > 3/4 cup.
Comment 3:
The authors need to address the inconsistency between their claim of using distribution-based quartiles for intake categories and their established serving size definitions. The manuscript uses fixed ½- and ¾-cup thresholds, but the Methods section defines “low” as less than ½ cup and “moderate” as ½ to ¾ cup. You should describe your method for determining cut points and explain the reasoning behind your selection. The text should eliminate “quartiles” since fixed servings were used but researchers must show their actual cut points if they applied quartiles in the study.
Response to comment 3:
Thank you for your comment focused on consistency of definitions. We updated the intake categories at lines 215-217 as follows: "we categorized pulses consumption based on the distribution intake within the population in serving sizes, where 1 serving size is 1/2 cup (113.4 grams)."
Comment 4:
Third, tighten internal counts. The research involved 6,958 participants who consumed pulses at least twice a week, with 4,584 participants in this group. The consumer-only models in Table 3 include 4,583 participants. Still, researchers need to confirm if one observation was excluded because of missing covariate data and include this exclusion information in the text. The text percentages require the correct denominators to calculate sex distribution percentages across intake categories in Table 1.
Response to comment 4:
We modified the analytic approach so that the models no longer use odds ratios but have calculated predicted probabilities. This change allowed us to include the non-consumers in the model and not have a reference group. We have updated the sample denominator in Table 3. Missing cases were due to incomplete data as reported on lines 239-240: "The multivariate logistic regression models included participants with non-missing data for pulses consumption, covariates, and outcome data." Our data is weighted, so the percentages shown in Table 1 are also weighted, including for sex. We added the word weighted in the tables titles to improve clarity.
Comment 5:
Fourth, prevent potential overadjustment. The AHEI-2010 scoring system includes nuts and legumes as a single scoring component. The inclusion of pulses within the “nuts and legumes” component of the AHEI-2010 scoring system might result in exposure-related variance being partially controlled through adjustment. The researchers should conduct a sensitivity analysis using an AHEI version that excludes nuts/legumes or select a different diet-quality score that excludes legumes. The Methods section requires an explanation about the AHEI version used for sensitivity tests, along with descriptions of any notable outcome variations.
Response to comment 5:
We included all four models tested and show these in the revised Table 3. Model 1 is unadjusted; model 2: is adjusted for covariates and energy intake; model 3 is adjusted for covariates, energy intake and diet quality with the AHEI measure; and model 4 is adjusted for covariates, energy intake, and diet quality using the measure derived from DASH. We included a model adjusted for diet quality using a score based on the DASH (Dietary Approaches to Stop Hypertension) diet that considers both macro- and micronutrients, rather than individual foods (such as nuts and legumes). We also added the Wald F and Wald F p-values for each covariate, which represent the impact of each covariate in the model. The associations remained very similar across the models.
Comment 6:
Fifth, report model diagnostics and trends. The researchers conducted survey-weighted logistic regression analysis and performed linear trend tests to evaluate ordered intake categories through median gram values for each group. The study should present an adjusted predicted prevalence or marginal probabilities graph that illustrates dose–response effects at various intake levels.
Response to comment 6:
Thank you for your comment. We modified our analysis for predicted marginal probabilities; changes in the methods can be found on lines 249-251: " We report the predicted marginal proportion for the results of these models for each level of pulses consumption and 95% confidence limits." We have revised Table 3, which now presents the dose-response effects of various levels of intake (none, low (<1/2 cup), moderate (≥1/2 cup to 3/4 cup), and high (>3/4 cup).
Comment 7:
The researchers need to check energy intake, AHEI, and pulse grams for acceptable multicollinearity and present variance inflation factors or make a brief statement about it.
Response to comment 7:
Thank you for the comment. We have added the following text to the methods section on lines 252-256. “The variance inflation factor (VIF) was used to assess multicollinearity between the model covariates of level of pulse consumption, energy intake, and, when applicable, diet measures. For the reported models, pulse consumption, energy intake and diet measure all had a VIF of less than 3. A VIF <3 indicates a low correlation between the variables in the model”
We also included energy intake and diet quality in our analysis of pulses intake for two reasons: 1) Energy intake as a measure of total caloric consumption is essential for interpreting overall dietary patterns. Pulses are nutrient-dense foods that contribute significantly to total caloric intake. Assessing the impact of pulses consumption on metabolic syndrome requires consideration of the total energy that they add to the diet, providing a more comprehensive understanding of their role. 2) Influence of diet quality as measured by AHEI on metabolic syndrome. Metabolic syndrome is affected by a variety of dietary factors, including sugar-sweetened beverages and dietary fat distribution. By evaluating diet quality, we can assess how the inclusion of pulses, which are known for their favorable nutrient profiles, enhances overall diet quality and contributes to improve metabolic outcomes. This holistic approach allows us to better understand the multifaceted impact of pulses consumption on metabolic syndrome. We also added the justification in the text at lines 243-252. In the manuscript Table 3, we show all models (unadjusted, adjusted for energy intake only, adjusted for energy and diet quality (AHEI), and adjusted for energy and diet quality (DASH)) to ensure no substantial differences were observed.
Comment 8:
The researchers need to extend their analysis of participants who consumed low amounts of food. The low consumer group showed the lowest diet quality and highest obesity and abdominal obesity rates despite their reported low calorie intake and high physical activity minutes according to descriptive tables which suggest possible measurement errors or confounding factors.
Response to comment 8:
Thank you for this comment. Extending analyses of this low-consumer group is beyond the scope of our manuscript, but the observation that they have lower diet quality and higher obesity prevalence raises questions for future research. In the discussion, we added: "Those who did not consume pulses had a higher intake of fruits, vegetables, and whole grains and a lower intake of alcohol compared to those who had a low consumption. These findings suggest that future studies are needed to explore dietary patterns with or without pulses associated with metabolic syndrome, as well as the longitudinal impact of different amounts of daily pulses consumption among Latino individuals on developing metabolic syndrome." found on lines 393-398.
Comment 9:
The paper requires additional details about energy reporting inaccuracies and NCI assessment methods for lifestyle behaviors to achieve better transparency. The study needs to conduct a sensitivity analysis that removes data to eliminate implausible energy reports.
Response to comment 9:
Thank you for your recommendations. We utilized the NCI method for measurement error and energy intake as noted in lines 152-154: "To account for dietary measurement error and estimate usual intake distribution, the National Cancer Institute (NCI) method was applied as described in previous publications". Dietary recalls with implausible energy reports were not included in our analytic sample, the description is noted in lines 115-117: "Of the 16,415 individuals at Visit 1, we excluded those with missing dietary information (n=86) and with implausible energy intake (defined as <500 or >6,000 kcal/day for females and <800 or >6,000 kcal/day for males; n=234)."
Comment 10:
The researchers need to organize their presentation of heritage-specific findings in a more structured way. The supplemental tables show that different heritage groups exhibit varying levels of connection between pulse consumption and metabolic syndrome risk, with particular groups showing more pronounced relationships.
Response to comment 10:
The Hispanic/Latino heritage groups are defined by the sampling strategy in HCHS/SOL and the reported heritage by individuals recruited to the study. We updated our interpretation of the results by heritage group which is now found on lines 366-373: "Analysis by heritage showed that adults of Cuban heritage who had a high consumption of pulses had statistically significantly lower metabolic syndrome prevalence (Predicted marginal model= 0.55, 95%CI = 0.50,0.59) than those who had a low consumption (Predicted marginal model= 0.77, 95%CI = 0.66,0.86). Central Americans differed showing a higher prevalence of metabolic syndrome at moderate pulses consumption, but the prevalence was not statistically significant. The other heritage groups reported lower prevalence of metabolic syndrome at moderate and high pulses consumption, but they were not statistically significant (Table S1)."
Comment 11:
The main text requires a short overview of cultural eating habits and details about whether researchers tested an interaction term (p-interaction). The study establishes its research boundaries and explains how variables relate to each other in the eighth step.
Response to comment 11:
We added a short overview of cultural eating habits for Hispanics/Latinos in lines 444-453 understanding that there is substantial heterogeneity in dietary habits. We also added text that we did exploratory analysis for Latino heritage groups but did not test for interaction terms in lines 256-258 as follows: "We also performed exploratory analysis for Latino heritage groups based on prior knowledge of differences by heritage groups."
Comment 12:
The Discussion section reveals that the study used a cross-sectional design, and participants answered survey questions based on self-reported information. The analysis contains two statements about metabolic syndrome patients changing their bean consumption habits and uncontrolled confounding factors that include food insecurity and cooking techniques. The research maintains a relationship between variables but avoids making any claims about causality.
Response to comment 12:
Thank you for your comment, but we want to clarify any potential misunderstandings. The research design being a cross-sectional analysis is not “revealed” in the Discussion but specified in the Methods. We emphasized this in the Discussion to avoid any implication of causality. Although dietary and behavioral data are based on self-report, the components that define metabolic syndrome were all measured during an in-person visit. We also did not measure food security or cooking techniques, and for that reason, we could not control for those variables. These factors may have influenced the individuals’ pulses consumption and are questions for future research. In the discussion we referenced that a previous study found that cooking methods could improve pulses consumption (Winham DM, Tisue ME, Palmer SM, Cichy KA, Shelley MC. Dry Bean Preferences and Attitudes among Midwest Hispanic and Non-Hispanic White Women. Nutrients. Jan 15 2019;11(1)doi:10.3390/nu11010178.)
Comment 13:
The tenth step requires researchers to enhance their reporting quality. The researchers need to detail their survey methods, confirm the use of weight measurements, strata, and PSU components, and describe their strategy for handling missing covariate data. The study needs to include 95% confidence intervals for all prevalence estimates in the text and complete p-values for instances where “<.001” appears in the journal's preferred format.
Response to comment 13:
Original and detailed survey methods, weight measures, strata and PSU components are described in detail in the cited reference in the material and methods study design and participants section (Gallo LC, Carlson JA, Sotres-Alvarez D, et al. The Hispanic Community Health Study/Study of Latinos Community and Surrounding Areas Study: sample, design, and procedures. Ann Epidemiol. Feb 2019;30:57-65. doi:10.1016/j.annepidem.2018.11.002). All HCHS/SOL analyses use this published methods paper to avoid lengthy repetitive description of methods for this epidmiological study as is standard practice. We added information on missing covariates in the statistical analysis section in lines 239-240: " The multivariate logistic regression models included participants with non-missing data for pulses consumption, covariates, and outcome data." We also added p-values for numbers <.001 in the tables.
Comment 14:
The abstract needs to present the complete category definitions and adjusted ORs that appear in the final version.
Response to comment 14:
Thank you. We adjusted the abstract according to the updated results that do not include adjusted Ors anymore.
Round 2
Reviewer 1 Report
Comments and Suggestions for Authors
Thank you, the improvement is huge, you went out of your way to improve.
Reviewer 2 Report
Comments and Suggestions for Authors
The authors have made sufficient improvements in the manuscript to meet my satisfaction. This paper will be an excellent addition to the journal.